# Maintained Spatial Learning and Memory Functions in Middle-Aged α9 Nicotinic Receptor Subunit Knock-Out Mice

**DOI:** 10.3390/brainsci13050794

**Published:** 2023-05-12

**Authors:** Sergio Vicencio-Jimenez, Paul H. Delano, Natalia Madrid, Gonzalo Terreros, Juan C. Maass, Carolina Delgado, Pascal Jorratt

**Affiliations:** 1Departamento de Neurociencia, Facultad de Medicina, Universidad de Chile, Santiago 8320328, Chile; 2Biomedical Neuroscience Institute, Facultad de Medicina, Universidad de Chile, Santiago 8320328, Chile; 3Otolaryngology Department, School of Medicine, Johns Hopkins University, Baltimore, MD 21231, USA; 4Department of Otolaryngology, Hospital Clínico Universidad de Chile, Santiago 8320328, Chile; 5Centro Avanzado de Ingeniería Eléctrica y Electrónica, AC3E, Universidad Técnica Federico Santa María, Valparaíso 2390136, Chile; 6Instituto de Ciencias de la Salud, Universidad de O’Higgins, Rancagua 2841935, Chile; 7Interdisciplinary Program of Physiology and Biophysics, Institute of Biomedical Sciences, Faculty of Medicine, Universidad de Chile, Santiago 8320328, Chile; 8National Institute of Mental Health, Topolová 748, 250 67 Klecany, Czech Republic; 9Third Faculty of Medicine, Charles University, Ruská 87, 100 00 Prague, Czech Republic

**Keywords:** auditory efferent, nicotinic receptor, cognitive impairment, spatial learning, novel object exploration

## Abstract

Age-related hearing loss is linked to cognitive impairment, but the mechanisms that relate to these conditions remain unclear. Evidence shows that the activation of medial olivocochlear (MOC) neurons delays cochlear aging and hearing loss. Consequently, the loss of MOC function may be related to cognitive impairment. The α9/α10 nicotinic receptor is the main target of cholinergic synapses between the MOC neurons and cochlear outer hair cells. Here, we explored spatial learning and memory performance in middle-aged wild-type (WT) and α9-nAChR subunit knock-out (KO) mice using the Barnes maze and measured auditory brainstem response (ABR) thresholds and the number of cochlear hair cells as a proxy of cochlear aging. Our results show non-significant spatial learning differences between WT and KO mice, but KO mice had a trend of increased latency to enter the escape box and freezing time. To test a possible reactivity to the escape box, we evaluated the novelty-induced behavior using an open field and found a tendency towards more freezing time in KO mice. There were no differences in memory, ABR threshold, or the number of cochlear hair cells. We suggest that the lack of α9-nAChR subunit alters novelty-induced behavior, but not spatial learning in middle-aged mice, by a non-cochlear mechanism.

## 1. Introduction

In 2022, at least 58 million people worldwide were estimated to be living with dementia, and due to the aging of the world population, this number will increase more than threefold by 2050 [1]. The estimated global burden of dementia in 2018 was USD 1 trillion, which is expected to rise to USD 2 trillion by 2030 [1]. Thus, understanding the causes that lead to dementia is an urgent public health issue. Among the risk factors that contribute to the development of dementia, hearing loss has recently emerged as one of the most relevant and, at the same time, least studied. Several epidemiological investigations have associated hearing loss with cognitive impairment [2,3,4] and dementia [4,5,6,7], and, in the newly proposed models for preventing dementia, hearing loss has emerged as one of the major modifiable risk factors for slowing cognitive decline in humans [6,7]. It is essential to consider that, among age-related conditions that are potentially modifiable and associated with dementia, hearing loss is the most widespread sensory disorder [8,9] and a leading cause of chronic disability in older adults [10]. Evidence shows that the level of hearing loss correlates significantly with the degree of cognitive dysfunction, in both demented and non-demented patients [5,11]. Moreover, this relationship cannot be explained solely by the effect that hearing loss may have on speech-based cognitive tasks [8], but also on other domains of cognition [11,12,13,14,15,16,17]. In addition, studies performed in animal models have provided behavioral and histopathological evidence linking hearing loss to cognitive impairment [18,19,20,21]. Furthermore, hearing impairment has been associated with hippocampal degeneration and spatial memory impairments, accelerating the onset of the phenotype of Alzheimer’s disease [19,20,21,22].

Growing evidence shows that auditory efferent pathways have a role in the development of age-related hearing loss [23,24,25], and in cognitive functions such as selective attention [26]. At the neuroanatomical level, the auditory efferent system is a neural network that originates in the auditory cortex and projects to the cochlea through olivocochlear fibers [27]. Importantly, these descending projections modulate outer hair cell (OHC) activity via medial olivocochlear neurons, which activate α9/α10 nicotinic receptors (nAChR) located in the OHCs of the cochlear receptor [28] and also in vestibular hair cells [29,30]. Our previous studies using α9-nAChR subunit knock-out (α9 KO) mice showed that α9/α10 nAChR aids in ignoring auditory distractors in a selective visual attention task [26] and affects motivation to seek reward [31]. However, whether other cognitive functions, such as spatial learning and memory, are also altered in the α9 KO mice is unknown. Here, we explored spatial learning and memory in middle-aged (12-month-old) wild-type and α9 KO mice to unravel potential mechanisms behind inner ear aging and cognitive decline. We found that middle-aged α9 KO mice did not have spatial learning or memory impairments in a Barnes maze, but a trend of increased freezing time and a reduced number of times entering the escape box, suggesting an altered novelty-induced behavior.

## 2. Materials and Methods

### 2.1. Animals and Housing

The experimental data were obtained from 16 WT and 16 KO male mice from our breeding colony (KO mice on the 129/SvEv backcrossed to the CBA/CaJ background and WT littermates) originally provided by Dr. Douglas Vetter from the University of Mississippi. Mice aged 11 to 13 months and weighing 28–34 g were included at the start of the behavioral protocols. The genotypes of each mouse were confirmed by PCR screening of genomic DNA extracted and purified from the tail before behavioral training. Mice were housed in pairs in polycarbonate cages (27.5 × 16.5 × 13.0 cm) with a 12 h light/dark cycle (lights on at 20:00 h) in a temperature-controlled room (22 ± 2 °C). Mice were given water and food ad libitum and were handled and weighed every day by the same researcher. All procedures were carried out following the Guidelines for the Care and Use of Laboratory Animals (publication number 86–23, National Institutes of Health, revised in 1996) and approved by the Animal Bioethics Committee (protocol CBA number 0728 FMUCH from the Faculty of Medicine, University of Chile). All behavioral procedures (Figure 1A) were performed blinded to the genotype of the mice.

### 2.2. Barnes Maze

The Barnes maze is commonly used to evaluate spatial learning and memory. The protocol was based on Sunyer et al. (2007) [32]. Briefly, mice were assessed on an acrylic circular maze with a diameter of 90 cm and 12 holes with a diameter of 7 cm (Figure 1B). The protocol consisted of three phases: adaptation, acquisition, and probe trials. All protocol stages were recorded on video and analyzed offline using the ANY-maze v4.70 (Stoelting^®^) software.

#### 2.2.1. Adaptation Stage

Animals were placed inside a dark plastic container (10 × 5 × 5 cm) and located in the center of the maze. Immediately afterward, lights (1000 lux) were turned on, and after 10 s, the container was lifted and removed from the maze. The researcher guided animals to the escape box; if animals did not enter, they were carefully introduced into it. Once animals were inside the escape box, lights were turned off, and animals were kept inside the escape box for two minutes.

#### 2.2.2. Acquisition Stage

The acquisition phase was carried out over four consecutive days. Animals performed four trials daily, separated by an inter-trial interval of 15 min. At the start of each acquisition trial, the maze was cleaned with a 70% ethanol solution, and then animals were positioned inside the dark plastic container in the center of the maze. The lights were turned on, and after 10 s, the container was removed. Then, the animal could freely explore for three minutes to find the escape box below the target hole while the rest of the holes were closed. Mice were able to reach the escape box through four visual cues (one on each enclosure wall). The trial ended when the animal entered the escape box or when a three-minute period had elapsed. If the animal entered the escape box, the lights were turned off, and the mouse remained inside the box for one minute. If the animal did not find the escape box before the end of the trial, the researcher guided it to the box and left it inside for one minute. The removal of the animal from the escape box marked the end of the trial, and the animal returned to its home box until the next trial. The behavioral measures evaluated were latency, distance, and number of errors (pokes into an incorrect hole) before reaching the escape box (total measures) and before the first encounter with the target hole (primary measures).

#### 2.2.3. Probe Trial

On day five (24 h after the last acquisition trial), a test trial was carried out, in which the target hole (where the escape box was located) was closed. Once again, the mouse was positioned inside the container in the center of the maze. The lights were turned on, and after 10 s, the container was removed, and the animal could freely explore the maze. After 90 s, it was removed from the maze. Finally, to evaluate long-term memory, a second probe trial, the same as the probe trial assessed on day 5, was carried out on day 12, without any training session between days 5 and 12.

### 2.3. Open Field Test

To evaluate novelty-induced exploration, we used an open-field test in the presence and absence of a novel object. The apparatus was a box made with opaque acrylic (40 × 40 × 22 cm) with a luminous intensity of 40 lux. Each mouse was placed in the center of the apparatus and allowed to explore it for five minutes. At the end of these five minutes, a novel object (3 × 3 × 10 cm glass parallelepiped) was placed in the center of the apparatus, and the animal could explore for another five minutes (Figure 1C). The spontaneous locomotor activity was monitored with a digital video camera. This test was recorded on video and analyzed offline using the ANY-maze v4.70 (Stoelting, Wood Dale, IL, USA) software by a blind researcher. The variables analyzed were the following: total distance traveled, distance traveled in the center, center entries, and freezing time.

### 2.4. Electrophysiology

Using auditory brainstem responses (ABR), we estimated the response threshold of mice to a 15 kHz pure tone. ABR signals were obtained through three subdermal needle electrodes: two inserted percutaneously into the left and right ear canal in the direction of the round window, and one electrode located in the midline of the animal skull. An experimenter blinded to the genotype determined the lowest tone intensity (dB SPL) that evoked an averaged response of 512 trials by visual inspection of ABR responses.

### 2.5. Immunostaining Analysis

Mice under isoflurane anesthesia were transcardially perfused with NaCl 0.9%, followed by 4% paraformaldehyde (PFA). Cochleae were fixed with 4% PFA overnight at room temperature (RT) and decalcified in 10% EDTA overnight at 4 °C. Cochleae were dissected into the basal, medial, and apical turn, washed with phosphate-buffered saline (PBS) three times for five minutes each, and then blocked with 10% goat serum and 0.2% Triton X-100 in PBS for 1 h at RT. The samples were incubated overnight with primary antibodies at 4 °C. After that, they were washed with PBS three times for five minutes by slowly shaking and then incubated with second antibodies for 2 h with slowly shaking in the dark at RT. The samples were washed with PBS one last time. Then, they were incubated with Hoechst 33,258 (Thermo Fisher Scientific, Waltham, MA, USA) for 15 min in the dark. Finally, the samples were mounted in anti-fade medium for fluorescence (VECTASHIELD^®^, Vector Laboratories, Newark, CA, USA) over microscope slides. The images were obtained using the BX61WI 230 DSU microscope and CellR 2.7 (Olympus^®^). The number of hair cells in the organ of Corti and their morphology were determined across the epithelium in image stacks using anti-myosin VIIa antibody and ImageJ software. The primary antibody used was anti-myosin VIIa polyclonal antibody (1:200, Proteus Bioscience, Ramona, CA, USA), and the secondary antibody used was a FITC-conjugated donkey antibody (1:1000, Jackson Laboratories, Bar Harbor, ME, USA).

### 2.6. Statistical Analysis

Regarding the Barnes maze, (i) the measures of spatial learning (latency, distance, and errors), freezing time, and number of pokes into the target hole were analyzed by two-way repeated-measures ANOVA (RM-ANOVA) with Bonferroni post hoc comparison tests; (ii) the comparison of the numbers of trials where WT and KO mice entered the escape box were evaluated using Fisher’s exact test; and (iii) the number of pokes in each hole and accuracy of pokes as a measure of short-term and long-term memory were evaluated using unpaired *t*-tests. The measures of novelty-induced behavior (entries and time spent in the center, and freezing time) and hair cell numbers were evaluated using a two-way ANOVA. The ABR threshold at 15 kHz was evaluated using an unpaired *t*-test. The normal distribution of the data was assessed by Shapiro–Wilk, and non-normally distributed data were transformed to [Log (X + 1)] to satisfy the requirements of the ANOVA model. Significant differences were considered for statistical tests with a *p*-value <0.05.

## 3. Results

### 3.1. Difference in Acquisition Stage

To assess spatial learning during the four days of the Barnes maze acquisition stage, we measured the latency, distance, and number of errors before reaching the escape box (total measures) and before the first encounter with the target hole (primary measures). In order to evaluate differences between these measures, a two-way repeated-measures ANOVA (RM-ANOVA) was performed between genotypes along the four days of the acquisition stage [genotypes: (WT, KO) × days (1–4)]. Figure 2 shows that the behavioral performance of WT (n = 16) and KO (n = 16) mice improved along the days in the acquisition stage since there was a decrease in total and primary measures (Figure 2A–C), indicating that mice learned to use the visual cues to escape the aversive stimulus (light at 1000 lux). Comparing both genotypes, there was a tendency for a difference in the total latency (WT: 94.7 ± 62.6 s (mean ± SD), KO: 122 ± 53.9 s), as KO mice tended (*p* = 0.06) to spend more time before reaching the escape box (Figure 2A). Non-significant differences were found in total distance, errors, and primary measures (Figure 2A–C). Furthermore, during the time spent before reaching the escape box, KO mice had a higher percentage of freezing time (two-way RM-ANOVA, F_(1,1)_ = 4.79, *p* = 0.04; WT: 22.9 ± 16.4%, KO: 35.7 ± 19.5%) (Figure 2D), and they had a higher number of pokes into the target hole (two-way RM-ANOVA, F_(1,1)_ = 6.77, *p* = 0.01; WT: 6.3 ± 7.4, KO: 11.8 ± 14.2) (Figure 2E). We found a significant interaction between genotype and days in the number of pokes into the target hole (two-way RM-ANOVA, F_(1,3)_ = 2.928, *p* = 0.04), followed by a Bonferroni post hoc test (t = −3.74, *p* = 0.02) showing a significant difference in day 4. Considering all the trials (4 trials × 4 days), KO mice had a higher percentage of trials without entering the escape box (Fisher’s exact test, *p* = 0.004) (Figure 2F).

### 3.2. No Difference between Genotypes in Memory

We also evaluated short- and long-term memory in the same group of animals. To test short-term memory, a probe trial was performed 24 h after the last day of the acquisition stage (day 5) in which all holes of the Barnes maze were closed. WT and KO mice had a preference for pokes in the hole where the escape box was located in the acquisition stage (target hole), since it was significantly higher than the number of pokes in all adjacent holes (Figure 3A). No differences were found comparing the number of pokes in WT and KO mice per hole. Moreover, there were no differences in the primary latency (WT: 26.7 ± 26.4, KO: 39.1 ± 33.4) (Figure 3B). To evaluate long-term memory, the probe trial was repeated seven days after the first probe trial (Day 12). No differences were found comparing the number of pokes per hole in WT and KO mice (Figure 3C) or in the primary latency (WT: 32.4 ± 31.3, KO: 25.5 ± 23.2) (Figure 3D). According to these results, WT and KO mice showed no differences in short- and long-term memory. As expected, both genotypes decreased visits to the target hole 7 days later.

### 3.3. Novelty-Induced Behavior

The tendency for increased total latency (but not primary latency), increased freezing time, and a higher percentage of trials without entering the escape box in KO mice during the acquisition stage could have been explained by a possible difference in the reactivity to the escape box. To test it, after the performance on the Barnes maze, we evaluated the novelty-induced behavior of a subset of WT (n = 12) and KO (n = 11) mice in an open field with and without a novel object for five minutes each. An increase in exploration is associated with an increase in the interaction of mice with the novel object, which was at the center of the open-field test. The introduction of the object produced a significant increase in the number of entries into the center (two-way ANOVA, F_(1,1)_ = 6.40, *p* = 0.02) (Figure 4A) and time spent in the center (two-way ANOVA, F_(1,1)_ = 32.10, *p* < 0.005) (Figure 4B) for WT and KO mice. Considering both periods, there was a tendency to have more freezing time in KO mice (two-way ANOVA, F_(1,1)_ = 3.649, *p* = 0.06) (Figure 4C).

### 3.4. ABR Threshold and Hair Cells Number

To study the possible link between the previous results and age-related hearing loss, we measured the ABR threshold at 15 KHz and the number of hair cells in the organ of Corti. Once mice finished the behavioral tasks, the ABR threshold was assessed using auditory brainstem responses to a 15 kHz tone burst. Then, the cochleae of mice were dissected to assess the number and morphology of the hair cells using Hoechst, a cell nucleus-specific dye, and antibodies against myosin VIIa, a hair cell marker (Figure 5A). Both the number of inner hair cells (IHC) in basal (in 100 µm; WT = 10.2 ± 0.8; KO = 10.4 ± 1.6), medial (WT = 11.5 ± 0.4; KO = 11.2 ± 1.4), and apical turn (10.6 ± 0.5; KO = 10.6 ± 0.6) (Figure 5B), and the number of OHC in basal (WT = 33.2 ± 1.6; KO = 34.7 ± 1.6), medial (WT = 36.6 ± 2.7; KO = 34.7 ± 4.3), and apical (WT = 32.5 ± 3.5; KO = 32.0 ± 1.7) (Figure 5C) were not different between genotypes. Finally, although there was an elevation in ABR thresholds that could be explained by the age of mice (12 months), there was a non-significant difference between WT (53.0 ± 10.3 dB SPL, n = 5) and KO mice (45.0 ± 19.1 dB SPL, n = 10) (Figure 5D). The smaller number of animals in this experiment compared to the number in the Barnes maze is because some of them died after the anesthesia injection, which is consistent with the age-related increase in mortality upon anesthesia with doses of ketamine/xylazine [33].

## 4. Discussion

In the present study, we explored spatial learning and memory as a measure of cognitive performance in middle-aged WT and α9 KO mice. We sought to describe the possible role of efferent cholinergic transmission to the inner ear in preventing cognitive impairment in middle-aged mice. Our results showed no differences between genotypes in primary measures of spatial learning, short- and long-term memory, ABR threshold, or the number of cochlear hair cells. However, KO mice had a trend of increased total latency, a significant increase in freezing time in the spatial learning protocol, and a higher percentage of trials without entering the escape box. There was also a trend of increased novelty-induced freezing in KO mice.

### 4.1. No Difference in Spatial Learning and Memory in α9 KO

The measures to assess spatial learning in the Barnes maze include latency, errors, and traveled distance. These parameters are measured when rodents locate the target hole for the first time (primary measures) or when rodents enter the escape box (total measures) [34]. Although studies using the Barnes maze have reported both measures, it has been suggested that primary measures are more informative than total measures to determine spatial abilities [35]. The differences between primary and total measures could be explained because mice sometimes run directly to the target hole using external cues, but instead of entering the escape box, they leave it to explore the maze [36]. Since the performance in the Barnes maze can be influenced by non-cognitive factors, such as exploratory locomotor activity [36], a hypothesis to explain why KO mice had a trend of increased total latency, but similar primary latency, might be that KO mice are less prone to explore novel environments. The trend of increased freezing time, a significantly higher number of pokes into the target hole, and a higher percentage of trials without entering the escape box in KO mice during the acquisition stage could suggest a fear of entering the escape box, since a decreased response to novelty exploration could result from an increment in fear of novelty [37]. During days 5 and 12, there were no differences between genotypes in the number of pokes per hole or primary latency, suggesting that short- and long-term memory are not altered in α9 KO mice. Interestingly, when the escape box was removed on both days, there was no difference in the freezing time.

The results from the novelty-induced behavior show an exploration in both genotypes since they had an increased number of entries and time spent in the center after the object introduction. However, KO mice had a trend of increased freezing time, suggesting a possible novelty-induced freezing, which is in the same line as the reactivity for novel objects induced by the escape box.

### 4.2. Altered Response to Stress and Motivation for Reward in KO Mice

Mohammadi et al. (2017) [38] found that α9 KO mice had an increased anxiety-like behavior and corticosterone plasma level compared to WT mice after a sub-chronic stress protocol of five days of movement restraint. Although the Barnes maze uses weak aversive stimulation (light) as a negative reinforcement to avoid the stress produced by strong aversive stimuli [32], it still elicits a stress response, as evidenced by an increase in plasma corticosterone after Barnes maze training [39]. Therefore, another possibility to explain the differences between genotypes during the acquisition stage is the development of increased anxiety-like behavior in α9 KO mice due to the endogenous stress response to the Barnes maze. Moreover, previous works have evidenced that the motivation for reward is also altered in α9 KO mice [31]. Mohammadi et al. (2017) [38] found that α9 KO mice displayed significantly lower reward-seeking behavior after a period of reward discounting, suggesting the development of anhedonia-like behavior evaluated using sucrose solution preference over water. Besides, in a previous study, we found that α9 KO mice had significantly fewer lever presses and perseverative responses to obtain pellet rewards during selective attention to visual stimuli [31]. Therefore, a reduced motivation could be another way to explain the reduced number of times that KO mice entered the escape box in the Barnes maze.

### 4.3. Auditory and Vestibular Function Related to Cognitive Impairment

Noise-induced hearing loss can impair spatial learning and memory, and this cognitive impairment is correlated with ABR thresholds [19,21]. Importantly, the auditory efferent system has been proposed to function as a protector against acoustic trauma [40,41,42]. Although our experiments were performed under exposure to environmental acoustic stimuli, the stimulation of olivocochlear fibers can slow cochlear aging even under these acoustic intensity levels [21,23]. In those studies, the cochlear function was evaluated through the measurements of the auditory thresholds and the number of IHC and OHC in the organ of Corti. In the present work, we found that there were no differences in the number of IHC and OHC of the cochlea or in ABR thresholds between genotypes. We found a mild deterioration of ABR thresholds in WT and KO mice, which is in accordance with the results of Lauer (2017) [43] in 11–13-month-old WT and KO mice and was not correlated with the average of total (Spearman correlation, n = 13, Rho = 0.262, *p* = 0.372) and primary latency (Spearman correlation, n = 13, Rho = 0.433, *p* = 0.132) during the 4-days of acquisition stage in the Barnes maze. However, beyond the ABR threshold alterations, noise exposure can produce hidden hearing loss (HHL), damage in the synapses between inner hair cells, and auditory-nerve neurons without an auditory threshold shift [44]. A recent study shows a negative correlation between α9/α10 nicotinic receptor activity and the HHL since a 100 dB SPL noise exposure for 1 h produced loss of ribbon synapses in KO mice, which is an indicator of cochlear synaptopathy, but an absence of synapses loss in mutated mice with an enhanced cholinergic activity [42]. In future studies, it would be interesting to evaluate if there is a relationship between cochlear synaptopathy and cognitive decline.

### 4.4. Limitations in the Behavioral Study of KO Mice

Apart from its expression in the inner ear, α9-nAChR has been found in the dorsal root ganglia [45], pituitary and adrenal glands [46,47], and lymphocytes [48], among others. In a recent study, Lykhmus et al. (2017) [49] found α9 RNA transcripts and a high α9-selective signal in ELISA in some brain areas, including the hippocampus. Although those last results are controversial [50], the essential role of the hippocampus in learning and novel object exploration could be another possible explanation for our results.

The present results have been obtained in WT and KO mice aged 12 months, which could be considered middle age, as mice can live up to 2 or 3 years [51]. In this line, the maintained spatial learning and memory functions in the KO mice could be indicating an important period during presbycusis development, showing that at these relatively early stages of the pathology, there are no cognitive alterations in the spatial domain.

## 5. Conclusions

Our study demonstrates that the functional absence of the α9-nAChR is not associated with a deficit in spatial learning and memory in middle-aged mice. However, the trend of increased freezing time in the acquisition stage in the Barnes maze and novel object exploration suggests novelty-induced impairment. Since the deletion of α9-nAChR did not alter the ABR threshold or number of cochlear hair cells, we suggest that the altered novelty-induced behavior in α9-KO middle-aged mice is due to a non-cochlear mechanism, such as novelty-induced freezing/anxiety or less motivation. However, an important caveat is that these altered mechanisms could be developed early in life by compensatory mechanisms of the cholinergic deficit in the α9-KO mice.

## Figures and Tables

**Figure 1 brainsci-13-00794-f001:**
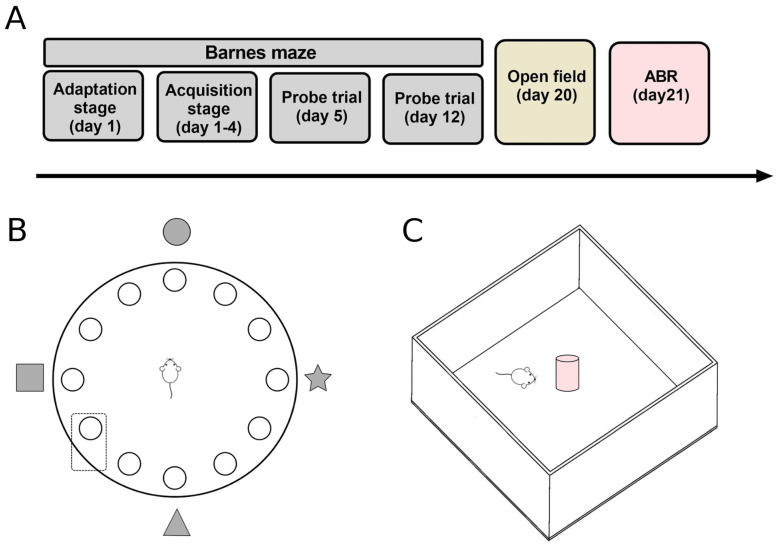
Behavioral apparatus and protocol. (**A**) Schematic description of the experimental timeline. (**B**) Illustration of the Barnes maze apparatus, a circular platform with 12 holes, one of which is connected to the escape box (inside the rectangle). Geometric figures represent the visual cues on the walls. (**C**) Diagram of the open field test in which animals explored the apparatus with a novel object for five minutes after a habituation period of five minutes without the novel object.

**Figure 2 brainsci-13-00794-f002:**
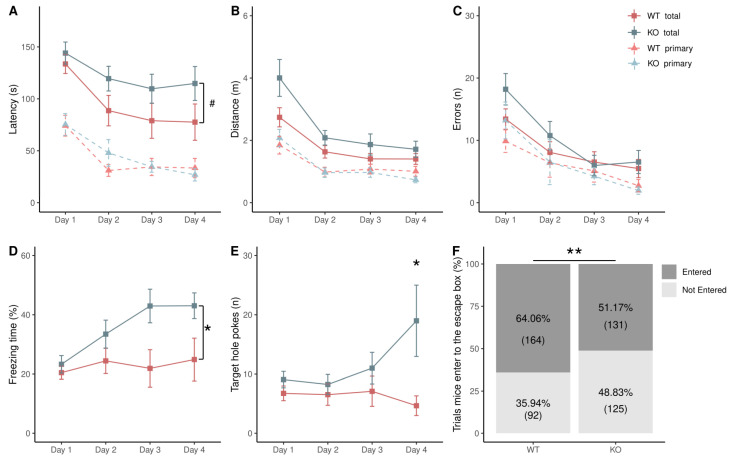
Spatial learning in middle-aged wild-type (WT) and α-9 nicotinic receptor knock-out (KO) mice. (**A**) Latency, (**B**) distance, and (**C**) errors during acquisition stage. (**D**) KO mice had an increased percentage of freezing time during the four days of spatial learning in Barnes maze. (**E**) KO mice had an increased number of target hole pokes on day four. (**F**) KO mice had a higher percentage of trials without entering the escape box (Fisher’s exact test). Data are shown in squares and triangles (mean ± SEM) for total and primary measures, respectively. WT (n = 16) and KO (n = 16) mice are shown in red and blue, respectively. Two-way RM-ANOVA followed by Bonferroni post hoc tests. * *p* < 0.05, ** *p* < 0.01, # *p* = 0.06.

**Figure 3 brainsci-13-00794-f003:**
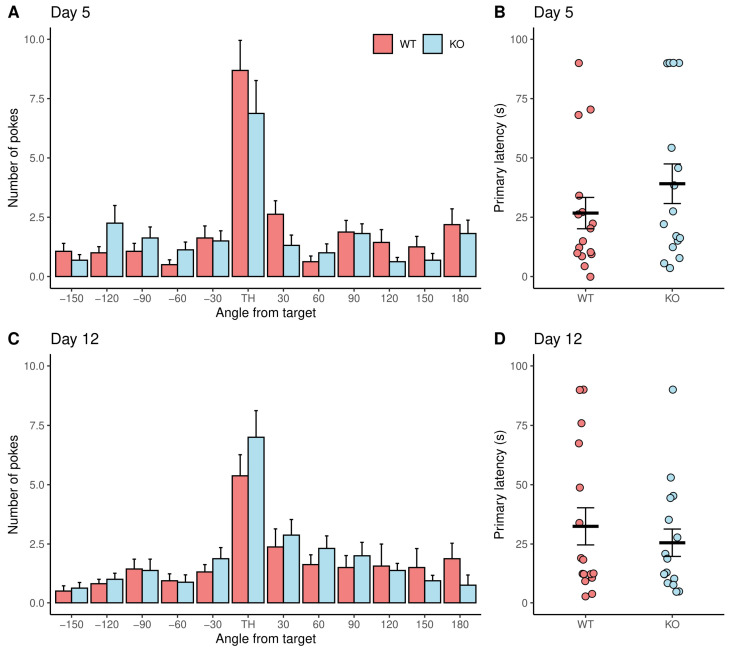
No differences in short- and long-term memory between wild-type (WT) and α-9 nicotinic receptor knock-out (KO) mice. (**A**–**D**) Number of pokes in each hole (**A**,**C**) and primary latency (**B**,**D**) during day 5 and day 12. Data are shown in red and blue bars/dots (mean ± SEM) for WT (n = 16) and KO (n = 16) mice, respectively. TH = target hole. Unpaired *t*-test.

**Figure 4 brainsci-13-00794-f004:**
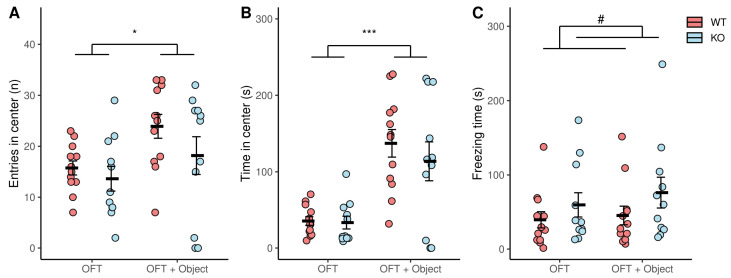
Novel object exploration in wild-type (WT) and α-9 nicotinic receptor knock-out (KO) mice. (**A**–**C**) The introduction of an object into the center of an open field test (OFT) produced an increase in (**A**) entries into the center and (**B**) time spent in the center for WT and KO mice. (**C**) Considering both periods, there was a tendency to have more freezing time in KO mice. Data are shown in red and blue dots (mean ± SEM) for WT (n = 12) and KO (n = 11) mice, respectively. Two-way ANOVA. * *p* < 0.05, *** *p* < 0.005, # *p* = 0.06.

**Figure 5 brainsci-13-00794-f005:**
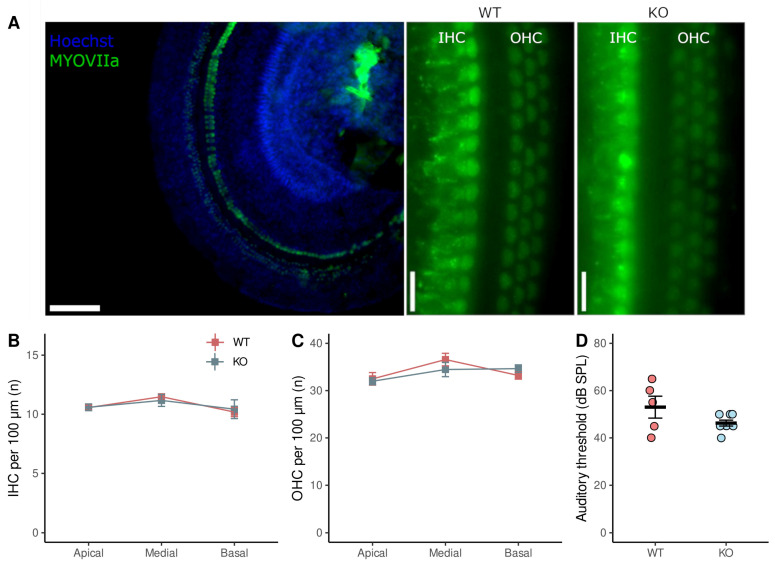
Number of cochlear hair cells and ABR thresholds in wild-type (WT) and knock-out (KO) mice. A stack of images across the epithelium was analyzed. (**A**) Inner hair cells (IHC) and outer hair cells (OHC) were stained with antibodies against myosin VIIa (MYOVIIa) and the nucleus with Hoechst. (**B**) Number of IHC and (**C**) OHC per 100 μm in apical, medial, and basal turn for WT (n = 9) and KO (n = 8) mice. (**D**) ABR thresholds for 15 kHz stimulus between WT (n = 5) and KO (n = 8) mice. Data are shown in red and blue squares/dots (mean ± SEM) for WT and KO mice, respectively. Scale bar = 300 and 10 μm.

## Data Availability

All data supporting described findings can be obtained upon reasonable request.

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
