# Peer review of "Maintained Spatial Learning and Memory Functions in Middle-Aged α9 Nicotinic Receptor Subunit Knock-Out Mice"

_brainsci, 2023, doi:10.3390/brainsci13050794_

Round 1
Reviewer 1 Report
This paper suggests that KO of α9 Nicotinic Receptor Subunit did not change spatial learning and memory function in mice. They used methods of Barnes maze/open field/ARB and IHC of cochlear.
It is useful for exploration of the mechanism/relationship between hearing loss and cognitive changes.
1. Authors guess that the behavior differences between genotypes were due to anxiety-like behavior in α9 KO mice. Why not compare the anxiety levels between genotypes in open field test? It is simple to do it.
2. Fig 5A only put one cochlear IHC of hair cells, it is better to put pictures of both WT and KO for comparison to show patterns, although there were no difference in numbers of hair cells.
3. Some sentences may need to be re-organized, such as line239-240
Some sentences may need to be re-organized, such as line239-240
Author Response
Comments from Reviewer #1
Comment 1. Authors guess that the behavior differences between genotypes were due to anxiety-like behavior in α9 KO mice. Why not compare the anxiety levels between genotypes in open field test? It is simple to do it.
Response: Thank you for this comment. Since novel object recognition involves first the exploration of the open field test without the object, the comparison of measurements as time and distance spent in the center during that period can be used to assess anxiety-like behavior. According to our results, there was no difference when comparing genotypes.
Comment 2. Fig 5A only put one cochlear IHC of hair cells, it is better to put pictures of both WT and KO for comparison to show patterns, although there were no difference in numbers of hair cells.
Response: We agree with this and have modified Fig.5A to include pictures for both WT and KO mice.
Comment 3. Some sentences may need to be re-organized, such as line239-240
Response: We appreciate this comment. We reorganized the suggested sentence (together with others) to clarify them.
Reviewer 2 Report
This is a review of “Maintained spatial learning and memory functions in middle-aged a9 nicotinic receptor subunit knock-out mice, submitted by Vicencio-Jimenez, et al. The alpha9 subunit of the nAChRs is a well-known contributor to the effects of olivocochlear activity on cochlear mechanics and, in general, hearing. Further, relatively recent reports have linked hearing loss with an increased propensity to, or acceleration of, dementia. Thus, the research described in this manuscript investigates whether a link exists between loss of alpha9 gene expression, and thereby loss of impact of cholinergic olivocochlear signaling on hearing, and cognitive abilities of spatial learning and memory.
The experiments were well designed and carried out. With very few minor exceptions (see below), the writing was excellent. There are only a few minor points the authors may wish to consider:
1) the background strain of the alpha9 KO mouse line should be identified in the Materials and Methods section, 2.1.
2) the authors describe their procedure for obtaining auditory thresholds. This was via ABR analyses at 15kHz. Typically, if a single stimulus is used to test ABR thresholds, a click will be used since it possesses many frequencies. Testing at one frequency with a single tone pip really isn’t testing “hearing”. It is only testing threshold at one of the more sensitive frequencies for mice. While in some regard, this procedure is not a fatal flaw, the authors should consider rethinking how to characterize this test as calling it a test for hearing is an overstatement of the test’s significance. Perhaps simply mentioning that mice were tested at a frequency they are normally quite sensitive to as a simple and expedient metric of the extent of potential age-related hearing loss (since ARHL typically starts at high frequencies and proceeds to lower frequencies) might be all that is required to ensure overstating the results does not occur.
3) at line 163-164, the authors should write something to the effect that the secondary antibody was tagged (or labeled) with FITC. As written, the sentence says that the label (FITC) was the secondary antibody.
4) there are a couple occurrences of double negative English constructs. For example, on line 197-198, the authors write that “Non-significant differences were found neither in total distance and errors nor in primary measures.” By what I see in the figure, I believe the authors mean that non-significant differences were detected. Simply eliminate “neither” to fix this issue 9of course, only if I am interpreting the authors’ meaning correctly of course). A similar issue is found in line 228.
5) there is a major gap/return that is out of place on line 222.
6) regarding hair cell counts assessed with the Hoechst counterstain, especially concerning OHCs, the authors should describe (simply, in a very short sentence) how they differentiated no loss of OHCs from potentially missing OHCs that have below them a Deiter’s cell nucleus. How were the Deiter’s cells and OHCs differentiated so we know that if an OHC were missing, the Detiter’s cell would not have been mistakenly counted as an OHC? The middle panel of Fig. 5A shows that the Hoechst labeling was not uniform, which further suggests that more information needs to be given as to how the different cells were identified and OHCs properly counted.
7) line 341- the authors describe cognitive impairment is correlated with ABR thresholds, but do not actually say this is correlated with decreased (presumably) thresholds. The sentence simply needs to be completed to fully convey the authors’ intent.
8) line 348-349: this is another place where there is a bit of overstatement of results associated with normal hearing based on a single frequency test. If a click was used, this would be more correct, but the authors should consider how to more fairly describe what can be concluded from their ABRs.
9) In the conclusions section, lines 377-379, regarding a potential for impaired exploration of novel environments/items, this seems to me not to be an age dependent outcome of the loss of alpha9. Likely this is a feature of the KO from very early on, given that the a9KO is constitutive. This brings up the limitation of all constitutive KO lines; that the mice are forced in unknowable ways to compensate for whatever process the null gene is normally involved with. I only mention this because it begins to slide away from the original premise of the project- to examine defects in middle/late life. In fact, these mice have had to (potentially!) develop strategies for living in their environment early in life that might leave them more prone to an anxious phenotype. Indeed, since the ACh-associated olivocochlear effect is one of dampening basilar membrane responses to sound, it is possible that the KO mice “perceive” all sounds as elevated (not by threshold changes, but by loss of “focusing” deflections along the basilar membrane, thereby altering recruitment of neighboring frequency regions) and this then leading to the anxiety state the authors’ reference.
a few minor comments can be found in the general comments to the authors
Author Response
Comments from Reviewer 2
Comment 1. the background strain of the alpha9 KO mouse line should be identified in the Materials and Methods section, 2.1.
Response: Thank you for this comment. The new version includes the background strain in Materials and Methods (Lines 79-80).
Comment 2. the authors describe their procedure for obtaining auditory thresholds. This was via ABR analyses at 15kHz. Typically, if a single stimulus is used to test ABR thresholds, a click will be used since it possesses many frequencies. Testing at one frequency with a single tone pip really isn’t testing “hearing”. It is only testing threshold at one of the more sensitive frequencies for mice. While in some regard, this procedure is not a fatal flaw, the authors should consider rethinking how to characterize this test as calling it a test for hearing is an overstatement of the test’s significance. Perhaps simply mentioning that mice were tested at a frequency they are normally quite sensitive to as a simple and expedient metric of the extent of potential age-related hearing loss (since ARHL typically starts at high frequencies and proceeds to lower frequencies) might be all that is required to ensure overstating the results does not occur.
Response: We agree with the reviewer. Therefore, we modifed the text to make explicit that only hearing sensitivity at 15Khz was evaluated as a quick measure to estimate age-associated changes in thresholds.
Comment 3. at line 163-164, the authors should write something to the effect that the secondary antibody was tagged (or labeled) with FITC. As written, the sentence says that the label (FITC) was the secondary antibody.
Response: Thank you for pointing this out. FITC specification was mentioned at line 165.
Comment 4. there are a couple occurrences of double negative English constructs. For example, on line 197-198, the authors write that “Non-significant differences were found neither in total distance and errors nor in primary measures.” By what I see in the figure, I believe the authors mean that non-significant differences were detected. Simply eliminate “neither” to fix this issue 9of course, only if I am interpreting the authors’ meaning correctly of course). A similar issue is found in line 228.
Response: We appreciate this comment. We agree that it was confusing and the sentences were modified in the suggested lines.
Comment 5. there is a major gap/return that is out of place on line 222.
Response: Thank you. The gap/return was fixed.
Comment 6. regarding hair cell counts assessed with the Hoechst counterstain, especially concerning OHCs, the authors should describe (simply, in a very short sentence) how they differentiated no loss of OHCs from potentially missing OHCs that have below them a Deiter’s cell nucleus. How were the Deiter’s cells and OHCs differentiated so we know that if an OHC were missing, the Detiter’s cell would not have been mistakenly counted as an OHC? The middle panel of Fig. 5A shows that the Hoechst labeling was not uniform, which further suggests that more information needs to be given as to how the different cells were identified and OHCs properly counted.
Response: Thank you very much for pointing this out. Cell counting was performed using MyoVIIa antibodies in a stack of several images and across the whole thickness of the epithelium. Although confirmed with Hoechst, MyoVIIa staining is sufficient for cell counting and it is one of the preferred method used for hair cells counting. The picture shown in the figure is just an image of the stack that is not perfectly aligned to the image plane. Thus, being “pseudoconfocal” the image losses brightness in the area that is not perfectly focused. We agree that the picture may be misleading. To avoid missinterpretation of the data, we modified the figure and used different images (also, we included one for wild type and one for knock out mice) to illustrate what we actually found in the study. We have also emphasized in methods (2.5. Immunostaining analysis) and in the legend (FIG5) that we checked a stack of images across the whole epithelium for cell counting and we used MyoVIIa for that purpose.
Comment 7. line 341- the authors describe cognitive impairment is correlated with ABR thresholds, but do not actually say this is correlated with decreased (presumably) thresholds. The sentence simply needs to be completed to fully convey the authors’ intent.
Response: We appreciate this comment. We included the values of the Spearman correlation between 15 kHz thresholds and average latencies in Barnez maze at lines 352-353.
Comment 8. line 348-349: this is another place where there is a bit of overstatement of results associated with normal hearing based on a single frequency test. If a click was used, this would be more correct, but the authors should consider how to more fairly describe what can be concluded from their ABRs.
Response: We agree with the reviewer. As we pointed out in the response to comment 2, we replaced "auditory threshold" by "ABR threshold at 15 kHz" in the revised manuscritpt.
Comment 9. In the conclusions section, lines 377-379, regarding a potential for impaired exploration of novel environments/items, this seems to me not to be an age dependent outcome of the loss of alpha9. Likely this is a feature of the KO from very early on, given that the a9KO is constitutive. This brings up the limitation of all constitutive KO lines; that the mice are forced in unknowable ways to compensate for whatever process the null gene is normally involved with. I only mention this because it begins to slide away from the original premise of the project- to examine defects in middle/late life. In fact, these mice have had to (potentially!) develop strategies for living in their environment early in life that might leave them more prone to an anxious phenotype. Indeed, since the ACh-associated olivocochlear effect is one of dampening basilar membrane responses to sound, it is possible that the KO mice “perceive” all sounds as elevated (not by threshold changes, but by loss of “focusing” deflections along the basilar membrane, thereby altering recruitment of neighboring frequency regions) and this then leading to the anxiety state the authors’ reference.
Response: We agree that we cannot discard that a9KO mice could have compensatory mechanisms in other brain circuits early in life. In the revised manuscript, we add a caveat to the reader about this point.